# Plasma Rich in Growth Factors in the Treatment of Endodontic Periapical Lesions in Adult Patients: 3-Dimensional Analysis Using Cone-Beam Computed Tomography on the Outcomes of Non-Surgical Endodontic Treatment Using A-PRF+ and Calcium Hydroxide: A Retrospective Cohort Study

**DOI:** 10.3390/jcm11206092

**Published:** 2022-10-16

**Authors:** Katarzyna Machut, Agata Żółtowska

**Affiliations:** Department of Conservative Dentistry, Faculty of Medicine, Medical University of Gdansk, 80-208 Gdansk, Poland

**Keywords:** advanced platelet rich fibrin plus, biomaterials, endodontics, root canal preparation, periapical lesion healing, cone-bean computed tomography (CBCT)

## Abstract

The study presents results of periapical lesion healing after one-visit root canal treatment (RCT) with Advanced Platelet Rich Fibrin plus (A-PRF+) application compared to a two-visit RCT with an inter-appointment calcium hydroxide filling. The comparison was made based on CBCT-Periapical Index (PAI) lesion volume changes and the occurrence of post endodontic pain. The results of 3D radiographic healing assessments based on volume reduction criteria were different from the CBCT-PAI. Based on volume changes, the healing assessment criteria-9 cases from the Study Group and six cases in the Control Group were defined as healed. Based on the CBCT-PAI healing assessment criteria, 8 cases from the Study Group and 9 cases from the Control Group were categorized as healed. The volumes of apical radiolucency were, on average, reduced by 85.93% in the Study Group and by 72.31% in the Control Group. Post-endodontic pain occurred more frequently in the Control than in the Study Group. The highest score of pain in the Study Group was five (moderate pain, n = 1), while in the Control Group, the highest score was eight (severe pain, n = 2). In the 6-month follow-up, CBCT scans showed a better healing tendency for patients in the Study Group.

## 1. Introduction

The periapical lesion, also known as symptomatic or asymptomatic apical periodontitis, is local inflammation that is caused by pulpal infection or traumas but can also be due to accidental over-instrumentation or chemical irritation of periapical tissues during root canal treatment [1]. The local inflammation is a result of disturbed stability between the infected root canal and host defense [2]. It may lead to the destruction of periodontal ligaments and to resorption of the mineralized tissue of both the tooth and/or alveolar bone. Mainly, periapical lesions are manifested as abscesses, granuloma, or cysts, which are radiolucent [3,4,5,6].

Several treatment methods have been described in the management of periapical lesions, such as: non-surgical root canal treatment (RCT), periradicular surgery, or tooth extraction. If possible, non-surgical approaches should be the treatment of choice. If the RCT is unsuccessful or impracticable, apicoectomy should be recommended. In the case that both the RCT and apicoectomy are ineffective, a tooth extraction would be suggested [7,8]. There are supplementary procedures that may probably enhance periapical lesion healing, such as: using a triple antibiotic paste (Hoshino paste containing Ciprofloxacin 500 mg, Metronidazole 400 mg, and Minocycline 100 mg) as a canal dressing, photo-activated disinfection (PAD) or magnetostimulation. However, there is still a need for further study in order to confirm their abilities [8,9]. RCT with a mineral trioxide aggregate (MTA) as an apical closure can be used for the apexification of an immature tooth with a large periapical lesion [10].

The main aim of apical periodontitis treatment is to remove the cause–eliminate bacteria from the root canal [11]. Mechanical debridement is not enough to eradicate microorganisms from a canal, so chemical irrigation with sodium hypochlorite 1–6% during and after instrumentation is recommended as it increases the effectiveness of bacteria reduction [12,13].

Non-hardening calcium hydroxide is used as an intracanal dressing with the highest activity during the first 2 weeks in two- or multi-visit RCT [14]. Due to an alkaline pH, it has a good proven eradication effectiveness of 80–100% of bacteria [15]. In addition, calcium hydroxide improves periapical tissue healing due to ionic effects: chemical dissociation into calcium and hydroxyl ions. Those ions have antimicrobial properties as they neutralize endotoxins produced by anaerobic bacteria. Calcium groups take part in bone tissue mineralization, while hydroxyl groups disintegrate the cytoplasmic membrane structure of bacteria cells and enhance tissue enzymes activity [16]. The main disadvantage of a two-visit treatment is a risk of the leakage of a temporary filling which may cause canal reinfection and impede the periapical lesions healing [17].

By definition, in a single-visit treatment, there is no risk of reinfection between the appointments. This approach additionally brings economical profits since some dental procedures, such as local anesthesia or the sterilization of tools, do not need to be repeated. Some studies suggested that a single-visit treatment is burdened with a higher risk of flare-ups, complications, or more frequent post-operative pain; however, the systematic reviews of Schwendicke et al. and Wong et al. did not confirm this statement arguing that there are no significant differences in a flare-up incidence between single and two-visit RCT [18,19].

Platelet-rich fibrin (PRF) is a second generation of autologous platelet concentrations (APCs). Except for platelets, PRF also contains other blood particles–B and T lymphocytes, monocytes, neutrophils, and stem cells, as well as growth factors (such as the platelet-derived epidermal growth factor (PD-EGF), platelet-derived growth factor (PDGF), vascular endothelial growth factor (VEGF), basic fibroblast growth factor (bFGF), and transforming growth factor-beta (TGF-β)) [20,21]. An important property of PRF is the prolonged release of growth factors at the application site, typically for more than 7 days [22]. Growth factors and released cytokines stimulate the activity of osteoblasts. The release of growth factors additionally expedites tissue regeneration by increasing the migration of fibroblasts [23,24]. PRF is commonly used in medicine and dentistry and has a proven ability to reduce postoperative pain and discomfort as well as accelerating recovery [25]. PRF seems to be particularly effective in regenerative dentistry as it is considered to enhance the formation of new bone tissue [26].

By decreasing the speed and shortening the time of the centrifugation of blood, a new modification of APCs was developed, and it is now known as advanced platelet rich fibrin plus (A-PRF+). A-PRF+ has regenerative abilities, which have been prove- to be higher than other formulations of PRF [27]. A-PRF+ also has a significantly increased level of released growth factors (TGF-β1, VEGF, PDGF, EGF, and IGF1) compared to the A-PRF and PRF. In addition, A-PRF+ promoted the enhanced migration and proliferation of human gingival fibroblasts [27,28]. Therefore, it may be speculated that the application of A-PRF+ in the apical region, before the final obturation of the root canal system, may accelerate the repair of periapical tissues.

The evaluation of a periradicular tissue condition is a vital element of the diagnosis, which allows the appropriate selection of a treatment method and an assessment of its results. The radiographic image of chronic apical periodontitis healing has been traditionally evaluated using the criteria known as the Periapical Index (PAI) [29] defined by Ørstavik et al. This scoring system uses a scale from one to five [Table 1] and is based on periapical radiographs. A score between one and two is recognized as healthy, and a score between three and five—as diseased. Other scales used to assess the outcome of periapical lesions healing after endodontic microsurgery based on periapical radiographs are Rud’s and Molven’s criteria [30]. All of these three scales are based on 2D imagery, which is a simplified representation of three-dimensional structures [30]. The overlapping of anatomical structures, geometric deformations, and background noise can thus impede the correct diagnosis [31,32].

Nowadays, cone-beam computed tomography (CBCT) is widely used in dentistry as an important tool in diagnostics and treatment planning [33]. Due to the cost and the exposure to radiation levels higher than in conventional radiographs, CBCT is still not commonly used for assessing root canal treatment outcomes [30]. According to the guidelines of the European Society of Endodontology (ESE), the use of CBCT should be considered after scrupulous clinical examination, including other tests such as conventional radiographs, in accordance to the ALARA (as low as reasonably achievable) principle [34]. The decision to use CBCT should be made individually when the periapical radiographs are not sufficient. In endodontics, CBCT must be adjusted to minimize the effective radiation dose by applying a high resolution and small FOV (field of view) (<5 cm) [34].

Based on clinical experience, CBCT is particularly recommended for diagnosing traumas as well as for the detection of periapical lesions and their differentiation between semi-solid granulomas and fluid-containing cavities (radicular cysts). This diagnostic tool is also valuable in identifying complex root canal systems, possible obliterations/resorptions, previous treatment complications/errors, and in planning periradicular surgery [35].

An index for periapical area evaluation was developed by Estrela C. et al. based on measurements of radiolucency interpreted on CBCT scans. It is known as CBCT–Periapical Index (CBCT-PAI). The CBCT-PAI is determined by the largest lesion extension in the 3 planes [36]. Table 2 presents the scoring scale of CBCT-PAI.

The aim of this study is to compare six-month follow-ups of periapical lesion healings after one-visit RCT with A-PRF+ application vs. two-visit RCT with inter-appointment calcium hydroxide dressings. The comparison will be made based on CBCT-PAI criteria, lesion volume changes, and the occurrence of PEP (post endodontic pain).

## 2. Materials and Methods

### 2.1. Case Selection

This retrospective cohort study was approved by the Independent Bioethics Committee for Scientific Research at the Medical University of Gdansk (No. NKBBN/607/2019) and was carried out in the Conservative Dentistry Clinic in the University Dental Center of the Medical University of Gdansk, Poland. The research was conducted between December 2019 and January 2022, and root canal treatments were performed by the same endodontist (K.M.). The primary sample size was determined by the limits imposed in the Bioethics Committee decision. Figure 1 shows, as a Consort 2010 Flow Chart, the distribution of participants and sample size at the time of the study. All RCTs were performed with a modified crown-down technique using nickel-titanium (NiTi) 0.04 rotary instruments (K3, Kerr, Glendora, CA, USA). The root canal irrigation protocol, along with manual-dynamic-activation (MDA) including: 5.25% sodium hypochlorite (NaOCl), 40% citric acid (CA), and distilled water.

Cases were selected using inclusion and exclusion criteria as described below.

The inclusion criteria were as follows:
Generally healthy patients, both sexes, aged from 20 to 50 years old.Root canal treatments of apical periodontitis cases with both preoperative and 6-month follow-up CBCT images.Teeth with appropriate amount of hard tooth tissue to be restored.Patients who had an intact restoration at follow-up.

The exclusion criteria were as follows:
Teeth with root fractures or perforations.Teeth which were previously endodontically treated.

In the study group (A-PRF+ group), after root canal preparation, blood was drawn from a median cubital vein and collected in glass tubes (10 mL). The blood was then centrifugated at 1200 rpm for 8 min in the Neuation iFuge D06 Premium Edition (Neuation Technologies Pvt., Gandhinagar, India) centrifuge to obtain advanced platelet-rich fibrin plus (A-PRF+), according to the technical documentation. The fibrin clot was squeezed between 2 sterile gauze pieces to remove the liquid and create a fibrin membrane. The prepared A-PRF+ membrane was then placed into the apex and pushed below the level of the cementodentinal junction by using Machtou hand pluggers-size 1/2 NiTi (red) and 3/4 (grey). Next, the canals were finally obturated and filled by a thermoplastic method (BeeFill 2in1 Obturation Kit, VDW GmBH, Munchen, Germany) with a calibrated gutta-percha cone and AH-plus sealer (Dentsply DeTrey GmbH, Philadelphia, PA, USA).

In the control group (Ca(OH)_2_ group), after root canal preparation, the canal was temporarily filled with a calcium hydroxide dressing for 2-weeks. After that period, the canals were finally filled using a thermoplastic method (BeeFill 2in1 Obturation Kit, VDW GmBH, Munchen, Germany) with a calibrated gutta-percha cone and AH-plus sealer (Dentsply DeTrey GmbH, Philadelphia, PA, USA).

The methods and materials are summarized in Table 3.

### 2.2. Pain Assessment

Patients were recalled 7 days after RCT. They were asked to indicate their perceived post-treatment pain on a horizontal visual analog scale (VAS). The values assigned on VAS were between 0 and 10. The description of pain intensity was presented to the patient by the examiner and was as follows: 0: No pain; 1–3: Mild pain; 4–6: Moderate pain; 7–10: Severe pain. Patients marked their post-operative pain levels in the presence of the clinician to ensure that they understood the instructions.

### 2.3. Healing Assessment

In the study, healing assessments were based on radiographic CBCT scans and CBCT-PAI criteria. Healing assessments were performed by two examiners (K.M. and A.Ż) who discussed discrepancies in the evaluation and reached an agreement in each case. The criteria of CBCT-PAI healing assessments [9] were as follows:

Favorable result:
Healed lesion: CBCT-PAI from 3, 4, 5 (+E/D) pre-operatively to CBCT-PAI 0, 1, 2 after 6-months.Healing lesion: CBCT-PAI from 3, 4, 5 (+E/D) pre-operatively improved, but is not 0, 1, 2 at a follow up CBCT.

Unfavorable result:
Diseased (not healed/healing): CBCT-PAI pre-operatively stays the same at follow up or is enlarged.

Another 3D radiographic healing assessment classification that is used in the literature and in evaluation after periradicular surgery is known as PENN 3D. Due to the lack of such classification for periapical lesion healing after non-surgical RCT, for this study these criteria were modified with the addition of volume changes. The criteria modified for RCT 3D with lesions volume reduction [31] are presented in Table 4.

### 2.4. Calculation of Lesion Volume in CBCT

The measurements of preoperative and postoperative volumes of the lesion in CBCT images were required to evaluate healing. The lesion volumes were calculated using the ITK-SNAP (free software under the GNU General Public License developed by the National Institutes of Health, the US National Institute of Biomedical Imaging and BioEngineering, the US National Library of Medicine, the Universities of Pennsylvania and North Carolina, and an independent developer group). Defect area segmentation and volume calculation were manipulated using the volumes at the highest resolution in CS 3D Imaging v3.5.18 Software (Carestream Health Inc., Trophy, Croissy-Beaubourg, France).

### 2.5. Statistical Analysis

The Mann-Whitney U and Kolmogorov-Smirnov non-parametric tests, as well as one-way ANOVA parametric tests were carried out to compare the healing and PEP assessment of the study and control groups. All statistical analyses were performed using SPSS Statistics v.28.0.1.1(14) (IBM, Armonk, NY, USA) with a significance level of *p* values < 0.05.

## 3. Results

A total of 40 teeth from 36 patients were evaluated six months after the endodontic treatment by using preoperative and recall CBCT scans. Cases are summarized in Table 5.

Both preoperative and postoperative lesion volumes were calculated using the free software ITK-SNAP version 3.8.0. The manual segmentation of the periapical defect was made in sagittal, axial and coronal views. Next, the defect was automatically reconstructed as a 3D object, and the lesions’ volumes were calculated. Examples of 3D reconstructions are shown in a Figure 2 and Figure 3, which present two lesions of one patient, and Figure 4 and Figure 5, which present two lesions from different groups with similar volume.

The volumes of apical radiolucency six months after endodontic treatment in the A-PRF group were, on average, reduced by 85.93% and in the control group by 72.31%. A statistically significant difference was confirmed in one-way ANOVA (*p* = 0.049). Volume reduction is presented in the box plot Figure 6.

Based on 3D Radiographic Healing Assessment Criteria in the Study Group, 9 teeth were categorized as completely healed, 8 teeth as being healed to a good standard, and 3 teeth as having only limited healing. In the Control Group 6 teeth were categorized as completely healed, 5 teeth as being healed to a good standard, 4 teeth as having only limited healing, and five teeth with uncertain healing. Results are presented in Table 6, and there was a statistically significant difference of *p* < 0.05. There was no incidence of any persistent pain, swelling, or fistula, and there was no reduction in periapical radiolucency in any of the cases, so none of the treatments were considered to have failed.

Based on CBCT-PAI healing assessment criteria in the Study Group, 8 cases were categorized as healed and, 12 cases were categorized as healing. In the Control Group, 9 cases were categorized as healed, 9 cases as healing, and 2 cases as not healed. The comparison of the post-operative vs. re-call CBCT-PAI’s in the Study and Control Groups are presented in Figure 7 and Figure 8 (Study and Control Groups respectively). Results are shown in Table 6.

To evaluate the incidence of post-endodontic pain, every patient from the Study and Control Group was invited to a re-call 1-week post root canal treatment. Post-endodontic pain occurred more frequently in the Control Group than in the Study Group (Mann-Whitney U test *p* = 0.221), but there was no statistically significant difference. The highest score in the Study Group was 5 (mild pain, n = 1), while in the Control Group, the highest score was 8 (severe pain, n = 2). Results are presented in Figure 9. In addition, after root canal treatment, 5 patients from the Control Group and 1 patient from the Study Group required additional analgesic therapy (Mann-Whitney U test *p* = 0.289), but there was no statistically significant difference. All of the patients were taking NSAID Ibuprofen for 2-days (400 mg^−1^ tablet/capsule every 4–6 h, but no more than 3 tablets/capsules in 24 h).

## 4. Discussion

The periapical area can be divided into two parts: (1) the inner—around the root apex and (2) the outer—surrounding the inner one [5]. The most frequently occurring microbes in the infected radicular system are Gram-negative anaerobes. Anaerobes produce endotoxin (aka lipopolysaccharide, LPS), which can activate the complement system by producing factor C5a [2,37]. This factor has an impact on generating chemotactic peptides. Due to chemotaxis in the inner part, the innate immune system cells–granulocytes (polymorphonuclear leukocytes, PMNs) and mainly the neutrophils prevail. Another known chemotactic factors for PMNs are interleukin-8 (IL-8), granulocyte chemotactic protein-2 (GCP-2), interleukin-1 (IL-1), interleukin-6 (IL-6), and interleukin-17 (IL-17) [5,38]. The first symptoms of inflammation are limited to hyperemia, edema, and the formation of exudate in the periodontal ligaments area [2,5]. It is interesting to note that in 95% of acute periapical periodontitis IL-8 was detectable in the exudative. This suggests that IL-8 may play a crucial role in the development of those lesions [39]. In the outer area of apical periodontitis, the macrophages and T lymphocytes are predominant. The migration of macrophages into the area of inflammation is much slower than PMNs [40]. Endotoxins stimulate macrophages to produce Il-1 and TNF-alpha. Those cytokines are pivotal in bone resorption.

Due to the decreased speed of centrifugation the total number of cells is higher in A-PRF+, compared to the standard PRF. An increase, especially in the number of neutrophils and lymphocytes was obtained [27]. All the cells were suspended in fibrin mesh, which stimulated the delayed resorption of the membrane and enabled typically 7–10 days lasting release of growth factors. A greater number of immune cells promote macrophage differentiation and maturation. The increase in the number of macrophages leads to an increase in the production of growth factors, which promotes the regeneration of bones and soft tissues [27]. In addition, the number of platelets is also higher, so the total amount of released growth factors (TGF-β1, VEGF, PDGF, EGF, and IGF1) is much higher in A-PRF+. A-PRF+ has the ability to induce angiogenesis and to act as a scaffold, which may accelerate the healing and regeneration of the damaged tissue [41]. PRF has proven its ability to enhance the proliferation of mesenchymal stem cells, such as periodontal ligament fibroblasts (PDLFs), dental pulp stem cells (DPSCs), human dental pulp cells (HDPCs), and human gingival fibroblasts, that play vital role in the regeneration of periapical periodontium [42]. Furthermore, PRF membranes had an antimicrobial effect, even against one of the main periopathogens of *Porphyromonas gingivalis* [42]. All the proven abilities of A-PRF+ in repair and the regeneration of hard and soft tissues seem to be beneficial as an additional key in RCT.

Moreover, there is evidence that calcium hydroxide has an application in endodontic regeneration [43]. Calcium hydroxide may induce the proliferation of mesenchymal stem cells, for example, PDLFs and DPSCs [43,44]. The exposure of DPSCs to calcium hydroxide induces their proliferation, differentiation to osteogenic cells, and also mineralization [44]. Alsalleeh et al. and Bhandi et al. studied present similar results, highlighting that both the concentration and time of exposure to calcium hydroxide contribute to its cytotoxic effect on cell viability [43,45]. In the Alsalleeh et al. study there is a suggestion that commercial calcium hydroxide preparations may be more toxic than calcium hydroxide and made ex tempore, especially Metapaste [43].

The healing of periapical lesions starts from the periphery, with size reduction caused by the new bone formation. The lesion becomes smaller in the radiographic image, and bone trabeculae with different radiopacities which fill the space of the lesion [46]. The majority of periapical lesion healing should be assessed at least between 6 [47] and 12-months [48] after the root canal treatment. Orstavik D. reported that half of cases present experienced advanced and complete healing at the 6-month visit, and after 12 months, 88% of these lesions were completely healed. The periapical lesion healing after RCT may take up to four years in some cases [48]. Prediction of RCT treatment success for teeth with periapical lesions is about 10–15% lower than for teeth treated with RCT for other reasons [49]. The reason for varying results among these studies may be: the extension of periapical lesions, the patients’ age, and some local or systemic factors that may interfere the healing of the lesion. Moreover, the using technique of mechanical debridement, appropriate irrigation, homogeneous root canal filling, but also good and sealed post-treatment restoration are crucial in resolving periapical lesions. The volume measurement of periapical lesions is one of the advantages of evaluating radiographic lesion healing processes using CBCT. In this study, after six months on the basis of 3D Radiographic Healing Assessments of volume changes, 9 cases from the Study Group and 6 cases from the Control Group were defined as healed (a total of 15). Based on the CBCT-PAI healing assessment criteria, 8 cases from the Study Group and 9 cases from the Control Group were categorized as healed (a total of 17). The difference between the results of these two classifications is due to the fact that a significant reduction of up to 90% in the periapical lesion volume may appear, even when one of the dimensions remains at a value that classifies as unfavorable in CBCT-PAI criteria.

CBCT is one of the most predictable methods of periapical radiolucency findings, with over 90% sensitivity and specificity [50]. Linear measurements of periapical lesions in three spatial planes on CBCT are more accurate than the measurements on conventional 2D radiographs [51]. The indicators to evaluate the periapical radiolucency advancement are: CBCT-PAI [36] or the CBCT-endodontic radiolucency index (CBCT-ERI), which measure the periodontal ligaments’ widths [52]. The above-mentioned scores in the available studies have the highest percentage of diversity between the observers [51]. Cotti et al. proposed the semiautomated segmentation of the volumes of the radiolucency on CBCT as a more accurate technique [51]. This conclusion seems to be corresponding with our results. Nevertheless, more studies have to be conducted, especially with the promising use of artificial intelligence detection of periapical lesions [51].

Post endodontic pain (PEP) has been reported in 25–40% of all endodontic cases. During the first 48 h after treatment, the pain reaches its greatest intensity and gradually decreases with time and typically lasts up to 7-days [53]. More than 50% of patients that suffer from PEP defined it as a severe pain [53,54]. The pain after the treatment is a main symptom of inflammation that may be caused by the extrusion of bacteria and dentinal or pulpal debris, but also irritants to the periapical tissues during root canal preparation [55].

Among all known preparation techniques, the least debris is extruded in the crown-down method [56].

In the Vishwanathaiah S. et al. review, in the majority of analyzed studies there were no significant differences in the occurrence of PEP between the single and multiple visit RCT [57]. One of the authors’ observations was that the studies conducted after April 2020 found no significant difference between the two procedures, in contrary to older studies [57]. The possible reason of this change is improvement of the root canal preparation and obturation technique, as well as the recognition of the importance of irrigation which has improved the quality of single visit treatment [57]. In the systemic review made by Wong et al., the induction of postoperative pain is the same in the single- and multiple-visit endodontic treatment. Both single and two-visit treatments showed similar healing and success rates [18]. Schwendicke and Göstemeyer in their studies, however, found that the single-visit treatment significantly increased the risk of flare-up (swelling), but they agreed that there is a similar incidence of postoperative pain or success rate, no matter what method is used [19].

Among the participants of this study, PEP occurred in 8 cases in the Control group and 5 cases in the Study Group. This study found no significant difference in the incidence of postoperative pain between the single-visit with the use of A-PRF+ and the two-visit treatment group (*p* > 0.05). Nevertheless, the intensity of pain was probably greater in the Control Group compared to the Study Group.

In the study, patients were assigned randomly to Study and Control Groups with block randomization methods to prevent selection bias and to ensure a balance in sample size across groups over time. This study reached the minimum number of necessary samples required to meet the desired statistical constraints to have a confidence level of 95% that the real value is within ±5% of the measured value. The power of the statistical tests used in this study was not high. A larger sample size would give more power in statistical analysis, but it is often difficult to achieve it in clinical practice, where the number of patients meeting the inclusion criteria is limited and where patients do not always attend re-call visits. A multicenter clinical trial might be ideal. Conducting research independently in several centers and a subsequent meta-analysis of results may be an alternative method of testing; however, the results of meta-analysis may be inaccurate due to different treatment protocols and methods of evaluation.

## 5. Conclusions

The results of 3D radiographic healing assessments of RCT using modified criteria were different from those based on CBCT-PAI criteria. In the 6-month follow-up, CBCT scans showed a better healing tendency in patients in the Study Group than in the Control Group. The volumes of apical radiolucency were, on average, reduced by 85.93% in the Study Group and by 72.31% in the Control Group.

## Figures and Tables

**Figure 1 jcm-11-06092-f001:**
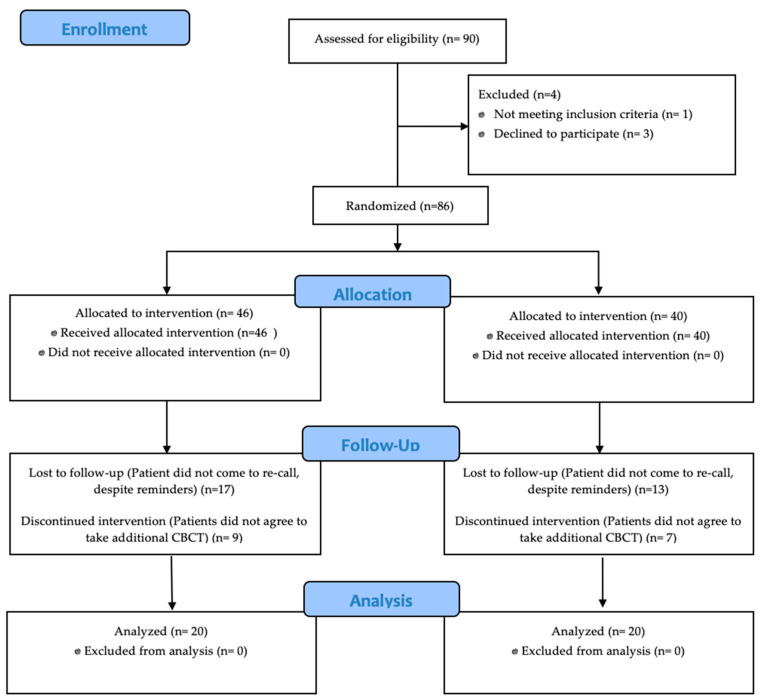
Consort 2010 Flow Chart.

**Figure 2 jcm-11-06092-f002:**
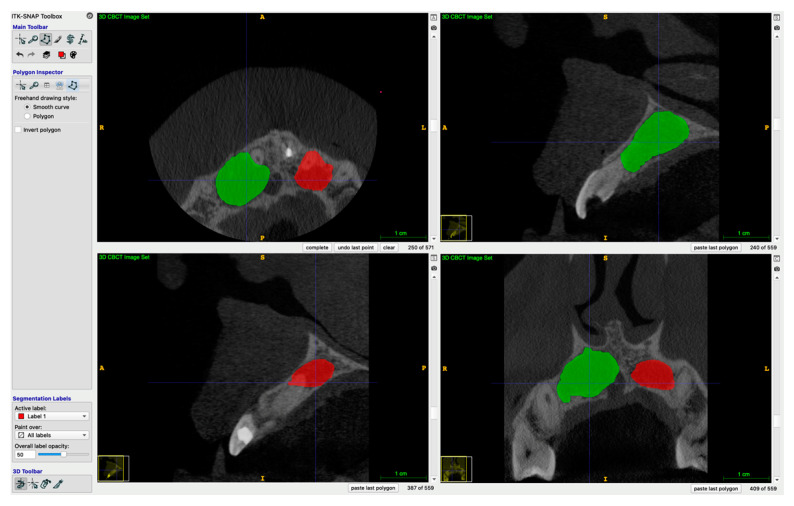
Example of a 3D reconstruction of two lesions pre-operatively. The lesion marked green was classified to the A-PRF+ group, and the one marked red was assigned to the control group.

**Figure 3 jcm-11-06092-f003:**
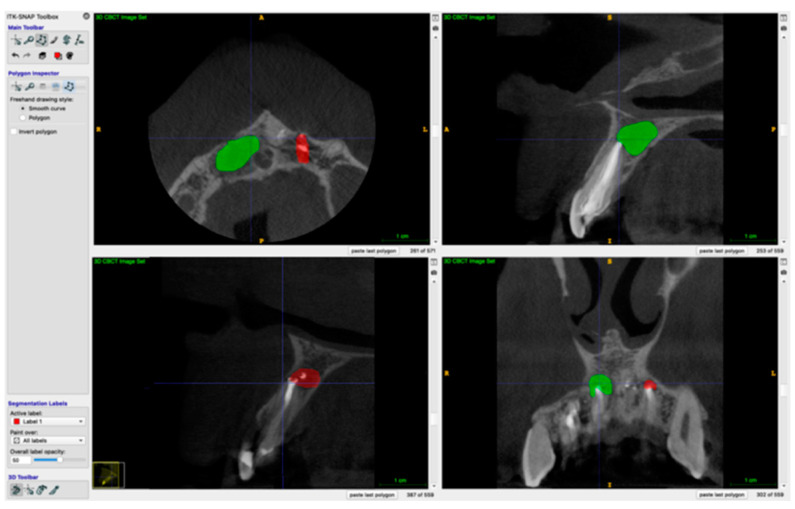
Example of a 3D reconstruction of two lesions at 6-month re-call.

**Figure 4 jcm-11-06092-f004:**
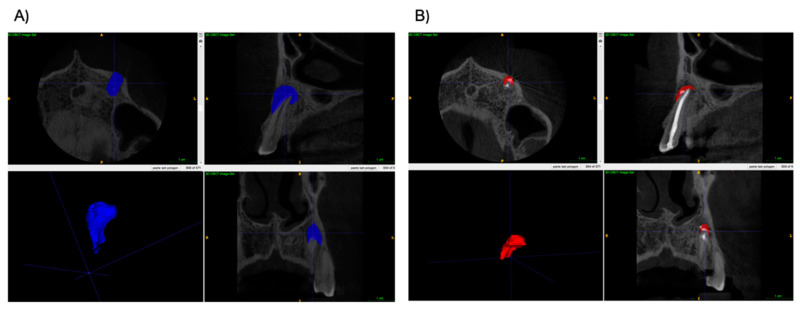
Example of a 3D reconstruction of lesions from the A-PRF+ group. (**A**) Reconstruction pre-operatively (Volume = 237.2 mm^3^), and (**B**) The lesion’s reconstruction at 6-month re-call. (Volume = 51.5 mm^3^).

**Figure 5 jcm-11-06092-f005:**
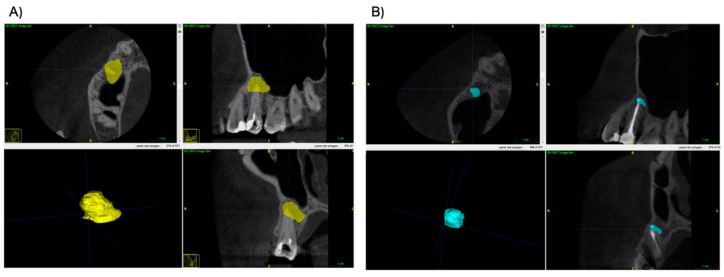
Example of a 3D reconstruction of lesions from the control group. (**A**) Reconstruction pre-operatively (Volume = 237.0 mm^3^), and (**B**) The lesion’s reconstruction at 6-month re-call. (Volume = 56.1 mm^3^).

**Figure 6 jcm-11-06092-f006:**
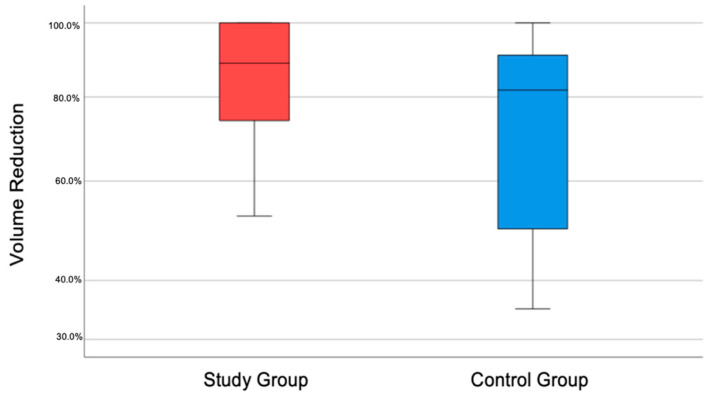
Lesion volume reduction box plot.

**Figure 7 jcm-11-06092-f007:**
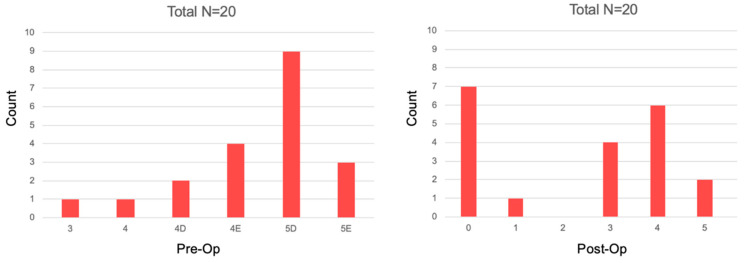
CBCT-PAI before and six months after the treatment in the Study Group.

**Figure 8 jcm-11-06092-f008:**
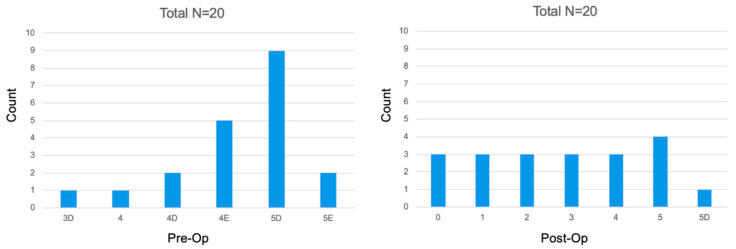
CBCT-PAI before and six months after the treatment in the Control Group.

**Figure 9 jcm-11-06092-f009:**
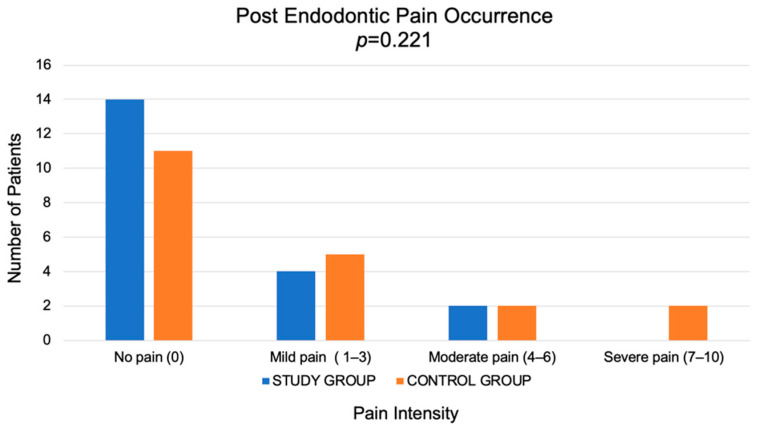
Results of post endodontic pain incidence in study and control groups.

**Table 1 jcm-11-06092-t001:** Periapical Index (PAI).

Periapical Index	
1	Normal periapical structures
2	Small changes in bone structures
3	Change in bone structure with mineral loss
4	Periodontitis with well-defined radiolucent area
5	Severe periodontitis with exacerbating features

**Table 2 jcm-11-06092-t002:** Periapical tissue evaluation: CBCT-PAI.

Score	Quantitative Radiolucency in Alveolar Bone Structures
0	Undisturbed periapical bone structures
1	Diameter of periapical radiolucency > 0.5–1 mm
2	Diameter of periapical radiolucency > 1–2 mm
3	Diameter of periapical radiolucency > 2–4 mm
4	Diameter of periapical radiolucency > 4–8 mm
5	Diameter of periapical radiolucency > 8 mm
Score (n) + E	Expansion of periapical cortical bone
Score (n) + D	Destruction of periapical cortical bone

**Table 3 jcm-11-06092-t003:** Summary of materials and methods.

Study Group	Control Group
All patients were treated by one investigator according to a standard regimen including elements of access, rubber dam, and establishment of asepsis. Local anesthesia (4% articaine with 1:100,000 epinephrine) was administered to the patient.	
Initial canal working length was established by using the electronic apex locator and a stainless-steel K-file (Endostar, Poldent LLC, Warsaw, Poland). Working length was confirmed by using radiographs. Canals were chemomechanically prepared with the modified crown-down technique using NiTi 0.04 rotary instruments (K3, Kerr, Glendora, CA, USA). Canals were irrigated with 5% sodium hypochlorite after each instrumentation cycle. All canals were irrigated with 40% citric acid (CA) for 1 min followed by a final irrigation with 5% sodium hypochlorite, and distillated water with the manual dynamic activation (MDA) and gutta-percha cone. Canals were then dried with sterile paper points.	
Application of A-PRF below the cementodentinal junction. Final obturation by the thermoplastic method with calibrated gutta-percha cone and AH-plus sealer, using the combination of a down-pack heat source with a Backfill extruder.	The canal was temporarily filled with non-hardening calcium hydroxide for 2 weeks. At the second appointment, the canal was obturated at the same appointment by using the thermoplastic method with a calibrated gutta-percha cone and AH-plus sealer, using the combination of a down-pack heat source with a Backfill extruder.

**Table 4 jcm-11-06092-t004:** 3D Radiographic Healing Assessment Criteria.

Score	Healing	Description
1	Complete	Small defect in bone surrounding the root apex- has widened the periodontal space up to 1 mm. Complete bone repair. Hard tissue surrounds the root apex The lesions volume reducing is from 90 to 100%.
2	Limited	A radiolucent area remains located around the apex and is connected with the periodontal space. Alveolar bone has not fully repaired. The lesions volume reducing is from 70 to 89%.
3	Uncertain	The volume of the low-density area appears decreased. The lesions volume is reducing from 50% to 69%.
4	Unsatisfactory	The volume of the radiolucency area is enlarged or unchanged or the volume reduce is lower than 50%.

**Table 5 jcm-11-06092-t005:** Case distribution of patients (n = 36) and treated teeth (n = 40) *.

			Study Group	Control Group
Sex	female		52.6% (n = 10)	45.0% (n = 9)
	male		47.4% (n = 9)	55.0% (n = 11)
Age (average) **			33.7	30.0
API (average) [%] ***			54.3	65.0
Tooth	maxillary	incisors	N = 10	N = 8
		canines	N = 2	N = 2
		premolars	N = 3	N = 4
		molars	N = 0	N = 0
	mandibular	incisors	N = 1	N = 1
		canines	N = 1	N = 0
		premolars	N = 1	N = 2
		molars	N = 2	N = 3

* Three patients fell into both groups (as each of them had one tooth treated with A-PRF+ and another tooth treated using a two-visit approach). In addition, one patient had two teeth treated simultaneously with the same method (A-PRF+). ** Participants’ age at the moment of RCT. *** API-Approximal Plaque Index is used to assess oral hygiene by determining plaque presence in the first and third quadrants lingually and in the second and fourth quadrants buccally.

**Table 6 jcm-11-06092-t006:** Results of 3D Radiographic Healing Assessment and CBCT-PAI healing assessment.

Healing Assessment	Study Group	Control Group
Completely healed	N = 9	N = 6
Good healing	N = 8	N = 5
Limited healing	N = 3	N = 4
Uncertain healing	N = 0	N = 5
Healed	N = 8	N = 9
Healing	N = 12	N = 9
Not Healed/No Healing	N = 0	N = 2

## Data Availability

All data are contained within the article.

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
