# Peer review of "Plasma Rich in Growth Factors in the Treatment of Endodontic Periapical Lesions in Adult Patients: 3-Dimensional Analysis Using Cone-Beam Computed Tomography on the Outcomes of Non-Surgical Endodontic Treatment Using A-PRF+ and Calcium Hydroxide: A Retrospective Cohort Study"

_jcm, 2022, doi:10.3390/jcm11206092_

Round 1

Reviewer 1 Report

This article describes a new approach to more effectively promote healing during non-surgical root canal treatment, using advanced platelet rich fibrin plus (A-PRF+) before a single-visit treatment, and compares this method with control group, which routinely uses calcium hydroxide dressing after root canal preparation for 2 weeks in two-visit RCT.

The authors use CBCT-PAI criteria, lesion volume changes and occurrence of PEP as evaluation criterias of results in the 6 months follow-up, and obtain the effect of A-PRF+ in RCT of adult patients with endodontic periapical lesions. The research has certain innovation and clinical application prospects, but there are a few points should to be clarified.

Major comments:

1. In this paper, the single-visit RCT with A-PRF+ was used as the study group, and the two-visit RCT with temporary filling of calcium hydroxide was used as the control group. There may be differences in the prognosis and risk between single-visit RCT and two-visit RCT, so is it necessary to add a group using two-visit RCT with A-PRF+, or a group using single-visit RCT without A-PRF+?

2. When comparing the differences between the healing and PEP assessment of the study group and the control group, the author can draft a table to display the results of statistical analysis,more clearly and intuitively showing the differences.

3. Are the severity of Endodontic periapical lesions consistent in the study and control groups? The severity of the selected cases in Figure 1 and Figure 2 is different, and two cases with the same severity of lesions maybe more suitable to explain the problem. At the same time, the images of the sagittal plane before and after treatment in the control group are the same.

Minor comments:

1. Please notice that when an abbreviation appears for the first time in the article, the full name of the abbreviation must be indicated first (line 127, PEP; Table 5, API).

Author Response

Dear Reviewer,

We would like to kindly thank you for the constructive comments you provided. Please see the attachment with our feedback.

Reviewer 2 Report

I read this article with great interest due to its innovative approach to this difficult topic which is endodontic treatment. As a reviewer I found a few points that require clarification.

Abstract.

Better not to use abbreviations in the abstract. If someone is not an expert in this topic, they may not read the entire text, if will have a problem with abstracting. And we all want our remedies to be quoted as often as possible  – RCT, A-PRF+

Introduction

Line 43

It would be good to give an example of what antibiotics are effective here in this clinical procedure

Line 97

Rud’s and Molven’s criteria.- please provide references

Line 106

According to the guidelines of European Society of Endodontology (ESE),- .- please provide references

Line 109

 ALARA- which means this abbreviation

Materials and methods

A good solution would be a graphical description of the study in the form of a flowchart

https://www.google.pl/search?q=description+of+the+study+in+the+form+of+a+flowchart&sxsrf=ALiCzsbj5aM4EYaVwThn09VHOHtgvPEJRA:1662720537457&source=lnms&tbm=isch&sa=X&ved=2ahUKEwj4svvXxIf6AhVJDewKHfp_A9EQ_AUoAXoECAEQAw&biw=1920&bih=969&dpr=1

Table 1

Initial canal working length is established by using the electronic apex locator and a stainless-steel

K-file. Working length is confirmed by using radiographs. Canals are chemomechanically

prepared with NiTi files.- please indicate the manufacturers and country of origin of the tools. This may affect the test result.

Line 150

centrifugated at 1200 rpm for 8 minutes- Has the centrifuge been checked in some way or is it working at this speed of rotation? Because it can also affect the obtained (A-PRF +). Has he been tested before in any of the other studies?

Discussion

Line 301

Alsalleeh et al. study suggests that propyleneglycol or barium  sulphate added to preparations of calcium hydroxide have cytotoxic effect on stem cells

Propylene glycol and barium sulfate are safe materials, I would say otherwise here. Please see this publication

https://pubmed.ncbi.nlm.nih.gov/27615202/

https://pubmed.ncbi.nlm.nih.gov/23656560/

 Good luck with your further research on this interesting topic

Author Response

(The authors gave the same response as above.)

Author Response

(The authors gave the same response as above.)
